

# Modularization in Belief-Desire-Intention agent programming and artifact-based environments

Gustavo Ortiz-Hernández[1], Alejandro Guerra-Hernández[2], Jomi F. Hübner[3] and Wulfrano Arturo Luna-Ramírez[4]

[1] Facultad de Ciencias Agrícolas, Universidad Veracruzana, Xalapa, Veracruz, Mexico
[2] Instituto de Investigaciones en Inteligencia Artificial, Universidad Veracruzana, Xalapa, Veracruz, Mexico
[3] Department of Automation and Systems, Federal University of Santa Catarina, Florianópolis, Brazil
[4] Departamento de Tecnologías de la Información, División de Ciencias de la Comunicación y Diseño, Universidad Autónoma Metropolitana-Cuajimalpa, Ciudad de México, México

## ABSTRACT

This article proposes an extension for the Agents and Artifacts meta-model to enable modularization. We adopt the Belief-Desire-Intention (BDI) model of agency to represent independent and reusable units of code by means of modules. The key idea behind our proposal is to take advantage of the syntactic notion of namespace, *i.e.*, a unique symbol identifier to organize a set of programming elements. On this basis, agents can decide in BDI terms which beliefs, goals, events, percepts and actions will be independently handled by a particular module. The practical feasibility of this approach is demonstrated by developing an auction scenario, where source code enhances scores of coupling, cohesion and complexity metrics, when compared against a non-modular version of the scenario. Our solution allows to address the name-collision issue, provides a use interface for modules that follows the information hiding principle, and promotes software engineering principles related to modularization such as reusability, extensibility and maintainability. Differently from others, our solution allows to encapsulate environment components into modules as it remains independent from a particular BDI agent-oriented programming language.

## INTRODUCTION

The increasing complexity of software demands tools and models allowing the development of such systems in a simpler way, while reducing the effort invested in maintaining and extending them. Several programming paradigms have been proposed with this purpose, *e.g.*, structured, declarative, object-oriented, and component-oriented programming. All of them have promoted engineering principles in response to the software requirements prevailing at their time; including maintainability, extensibility and reusability, among others. These principles have became desiderata when designing and developing software systems (*Abran et al., 2001*).

Corresponding author
Wulfrano Arturo Luna-Ramírez,
wluna@correo.cua.uam.mx

However, new software requirements have arisen during the so-called Internet revolution, demanding systems able to exhibit some degree of autonomy, reactivity and pro-activity for facing highly dynamic, distributed and heterogeneous scenarios. The agent oriented programming (AOP) paradigm (*Shoham, 1993*) copes naturally with such requirements, reducing user intervention (*Maamar & Moulin, 1997*) and promoting famed methods for facing complexity: decomposition, abstraction and hierarchy (*Jennings, 1999*; *Cuesta, Gomez & Gonzalez, 2008*; *Federico Bergenti & Zambonelli, 2004*). Additionally, the agents and artifacts (A&A) meta-model (*Weyns, Omicini & Odell, 2007*; *Ricci, Piunti & Viroli, 2011*) promotes a modular approach for programming the environment where agents are situated, providing a suitable model to externalize (*Ricci, Piunti & Viroli, 2009*) the functionality required by agents to act and perceive. Unfortunately, current AOP proposals usually address this complexity taking the multi-agent system (MAS) as a whole, disregarding the individual agent level, despite an agent is intrinsically a complex system as well. Therefore, we state that the strategies for dealing with complexity must be enforced also when designing and developing individual agents.

Modularization can significantly facilitate and improve such vision. Its adoption in the context of BDI-AOP has been widely discussed and developed in the literature (*Zanbar & Kaminka, 2019*; *da Rocha Costa, 2018*; *Ricci et al., 2019*; *Aschermann, Kraus & Müller, 2017*; *Busetta et al., 1999*; *Braubach, Pokahr & Lamersdorf, 2006*; *Dastani & Steunebrink, 2009*; *Cap, Dastani & Harbers, 2011*; *Dastani, Mol & Steunebrink, 2009*; *Madden & Logan, 2010*; *Ortiz-Hernandez, Guerra-Hernandez & Hoyos-Rivera, 2013*; *Hindriks, 2008*; *Van Riemsdijk et al., 2006*; *Ricci, Piunti & Viroli, 2009*). As stated by *Suryanarayana, Samarthyam & Sharma (2015)*, modularization can be understood as "*the logical partitioning of a software design so that the design becomes easy to understand and maintain*". Accordingly, the process of modularization must be achieved by reducing the degree of interaction and dependency between modules(coupling), while increasing the level of coherence between the elements composing a module (cohesion); in order to help maintaining tasks such as extension and scalability. Based on that, we hold that encapsulation (information hiding) enables high cohesion, and a well-defined interface allows low coupling. Then by solving encapsulation and interface we pursue modularization in such terms.

This article proposes an extension of the A&A meta-model, enabling modularization at the agent level. In this context, the Belief-Desire-Intention (BDI) agency model is adopted. Therefore, modules are intended to encapsulate functionality into independent and reusable units of code which can be dynamically loaded by agents. We identify four major issues need to be faced to support modularization of BDI agents in such terms: (i) Avoiding name-collision, (ii) satisfying the principle of information hiding, (iii) interfacing modules, and (iv) integrating environment components. We present a generic proposal, suitable for agent oriented programming languages adopting the agents and artifacts meta-model; and also a concrete implementation in JaCa–an integration of Jason and CArtAgO, an AOP language and an infrastructure for programming artifact-based environments, respectively.

The main idea of our approach is to take advantage from *namespace*, a syntactic notion to address these four issues. Thus, every component of a module has an associated namespace which makes it possible to resolve name-collision issues. Namespaces can

be private or public, so that components can be encapsulated to fulfill the information hiding principle. Module components may be associated with an abstract namespace to be concretized at runtime, once the module has been loaded. An abstract namespace notion provides both a bidirectional use interface for modules and a mechanism to load a module as many times as needed in different concrete namespaces, enabling a behavior close to that of the class/instance relationship proper to object-oriented programming (OOP). Adding namespaces to the bridge, *i.e.,* the component that mediates between agents and artifacts by translating operations and properties to actions and events, respectively, results in a modules system which transparently integrates environment components for both AOP languages and the artifacts infrastructure.

Our proposal is based on the approach followed in *Ortiz-Hernández et al. (2016)*. The main novelty is that beyond composing modules with only elements inherent to agent programs, *e.g.*, beliefs, objectives, intentions, we also consider components related to the environment, such as perceptions and actions. By including the environment components in the definition of module, agents can decide the particular set of perceptions and actions that will be independently handled by modules. In other words, a module handles only the actions and perceptions related to its functionality while the rest remain transparent.

This article is organized as follows: Given that modularization has been actively discussed in AOP, 'Related Work' reviews and compares different approaches reported in the literature and our current proposal. We intend such a review to be useful for establishing the contribution and novelty of our approach (encapsulating environment components into modules). 'The A&A Meta-Model' gives a description of the A&A meta-model, focusing on the elements involved in modularization.

For the sake of conciseness and clarity, we assumed *AgentSpeak(L)* (*Rao, 1996*) as the model of agency, but the ideas presented in this article can be generalized to other BDI programming languages.

Our proposal for introducing modularization in the A&A meta-model is exposed in 'Modularization in the Agents and Artifacts Meta-Model'. To prove how feasible our approach is, an implementation has been done in JaCa, the integration of Jason (*Bordini, Hübner & Wooldridge, 2007*) and CArtAgO (*Ricci et al., 2009*). This implementation is briefly discussed in 'Implementation'. Then, a case study inspired on an auction scenario for building a house is introduced in 'Case Study'. The case study is positively evaluated by coupling, cohesion and the Halstead complexity metrics (*Halstead, 1977*), when compared against a non-modular solution. 'Evaluation' offers the details of the evaluation. Finally, 'Discussion' discusses the suitability of the proposed approach to construct association, composition, generalization and cardinality relationships among modules (*Nunes, 2014*), so that it is possible to extend and build complex functionalities while promoting code reuse. It also mentions the way some OOP concepts, such as multi-inheritance, can be modeled using our approach. Finally, our conclusions and future work are offered in 'Conclusion and Future Work'.

# RELATED WORK

The diversity of approaches for adopting modularization in the BDI model of agency reveals the relevance of this subject for the AOP community. Below is a summary of works that propose various strategies to deal with the problems related to modularization in BDI-AOP.

*Busetta et al. (1999)* use modules to encapsulate beliefs, plans and goals related to a specific functionally, within a shared scope known as capability. Based on that, programmers indicate through scoping rules which elements can be accessed by other capabilities. A concrete implementation in JACK (*Howden et al., 2001*) is provided. Further, *Braubach, Pokahr & Lamersdorf (2006)* extend the concept of capability to ensure that all elements remain hidden from outside and that they form part of only one capability, assuring the fulfilment of the information hiding principle by preventing it for being violated. They provide a mechanism for configuring capabilities with some initial mental state, and an implementation in JADEX (*Braubach, Pokahr & Lamersdorf, 2005*) is also available. Both approaches propound an interface based on an explicit import/export statement as part of the capability header.

An association between modules and a specific goal is proposed by *Van Riemsdijk et al. (2006)*. In this proposal, goals are intended to be handled by a particular module. In other words, modules are constructed for being executed just for achieving those goals they are associated with. Within a module, every plan is executed one-by-one till either the goal is achieved or all plans has been tried. This work is conceived in the 3APL (*Dastani et al., 2004*) context.

*Hindriks (2008)* use an approach inspired in policy-based intentions, as described for GOAL (*Hindriks, 2009*). A condition representing a specific agent's mental state is associated with modules. When this condition is met, the module associated with it turns into the focus of the execution, while dismissing any other goal. The use interface of modules is given by the condition that triggers the activation, and which is defined declaratively. This strategy focuses on avoiding the pursuit of contradictory goals by means of the isolation of events and goals.

*Dastani & Steunebrink (2009)* propose modules as independent mental states on which agents can start reasoning, one at the time until a predefined condition holds. Such mental states are instantiated, executed, tested and updated applying a set of predefined operations. In other words, once the agent executes a module, control is transferred and returns to the agent's main program until the module ends its execution. These concepts are introduced in the context of 2APL (*Dastani, 2008*) by extending its operational semantics. Its corresponding implementation is also described by *Dastani, Mol & Steunebrink (2009)*.

*Madden & Logan (2010)* encapsulate capabilities through modularization. Their approach follows the XML's definition of namespaces (*Bray et al., 2006*). Each module is treated as a unique and separate namespace attached to a URI to be clearly identified. A distinct local goals-base, belief-base and events-queue are instantiated per module, then it is possible for programmers to specify, using an explicit export/import statement, the particular goals, beliefs and events that will remain visible and accessible to other modules. In this modules system, a module can be instantiated only once, therefore

references and changes to the exported part of the module are visible and accessible to every other module that imports it.

In *Cap, Dastani & Harbers (2011)*, an extension of *Dastani & Steunebrink (2009)* is presented to mainly enhance the interface by incorporating the concept of sharing scopes. They allow modules posting events, so they are available to other modules within a common scope. Initially, each agent contains one predefined sharing scope to contain the agent's main module instance. Then modules are activated by means of 'inclusion', *i.e.,* the module instance is added to the existing sharing scope; and deactivated by 'seclusion', that is to create a new sharing scope for adding the module.

*Ortiz-Hernandez, Guerra-Hernandez & Hoyos-Rivera (2013)* deal with modularization challenges by defining modules in the context of AOP, a set of beliefs and plans, but with the inclusion of a header with an import/export list referring to module components. Then, name-collision problem is tackled by using annotation mechanism to identify the module to which each belief and plan belongs. They also define a common and unique initial-module which components are accessible to every other module. A library for supporting their approach is available for Jason (*Bordini, Hübner & Wooldridge, 2007*)

These approaches are compared in Table 1. All of them provide diverse interfaces among modules and solve the name collision problem (NameCol), providing an instrument to manage the visibility of goals and events. However, some of them fails to fulfill the information hiding principle (InfoHid). Excepting our current and previous proposals, all the reviewed approaches are not independent (Indep) of the AOP language (Lang) used to conceive and implement them, probably preventing its reuse in other languages. Except for our current proposal, all of them miss a suitable mechanism to consider environment related components (Env).

## THE A&A META-MODEL

The A&A meta-model (*Weyns, Omicini & Odell, 2007*; *Ricci, Piunti & Viroli, 2011*) promotes a view of MAS where both the agents and the environment are first-class abstractions. Figure 1 introduces the meta-model. Such a view conceives environment as a set of artifacts that the agents can create, focus and dispose for pursuing their goals. In the A&A meta-model a MAS is modeled as a tuple $\langle Ags, Bdg, Env \rangle$, composed of a set of agents ($Ags$), an environment ($Env$), and a bridge ($Bdg$) between them. A full list of symbols used in the A&A model and in this work is shown in Table 2.

Given the generic nature of the meta-model, different AOP languages and artifact infrastructures can be adopted as shown in Fig. 2. The artifact infrastructure provides an abstraction of the environment that works as an agent interface among each other and their real environment. Agents and artifacts interact in BDI terms: Artifacts are the source of some of the beliefs of the agents, and the actions executed by the agents are indeed artifact's operations. Artifacts and environment interact in more traditional terms, *e.g.,* OOP. Any object can be encapsulated in an artifact, contributing in this way to what *Shoham (1993)* called "agentification", *i.e.,* the transformation of an arbitrary device into a programmable agent. The bridge is an *ad hoc* component that mediates between the agents

**Table 1   Different approaches for agents modularization in the BDI model of agency, compared regarding: their implementation language (Lang); their independence (Indep) with respect to the implementation language; the integration of environment components (Env); the fulfilling of the information hiding principle (InfoHid); the resolution of name collisions (NameCol); and the interface provided for the modules.**

| Modularization approach | Lang | Indep | Env | InfoHid | NameCol | Interface |
|---|---|---|---|---|---|---|
| *Busetta et al. (1999)* | JACK | × | × | ✓ | ✓ | Explicit import/export |
| *Braubach, Pokahr & Lamersdorf (2006)* | JADEX | × | × | ✓ | ✓ | Explicit import/export |
| *Van Riemsdijk et al. (2006)* | 3APL | × | × | × | ✓ | Goal dispatching |
| *Hindriks (2008)* | GOAL | × | × | × | ✓ | Mental-state condition |
| *Dastani & Steunebrink (2009)* | 2APL | × | × | ✓ | ✓ | Set of predefined operations |
| *Madden & Logan (2010)* | Jason+ | × | × | ✓ | ✓ | Explicit import/export |
| *Cap, Dastani & Harbers (2011)* | 2APL | × | × | ✓ | ✓ | Sharing scopes |
| *Ortiz-Hernandez, Guerra-Hernandez & Hoyos-Rivera (2013)* | Jason | × | × | × | ✓ | Unique-common scope |
| *Ortiz-Hernández et al. (2016)* | Jason | ✓ | × | ✓ | ✓ | Global/abstract namespaces |
| Our current proposal | JaCa | ✓ | ✓ | ✓ | ✓ | Global/abstract namespaces |

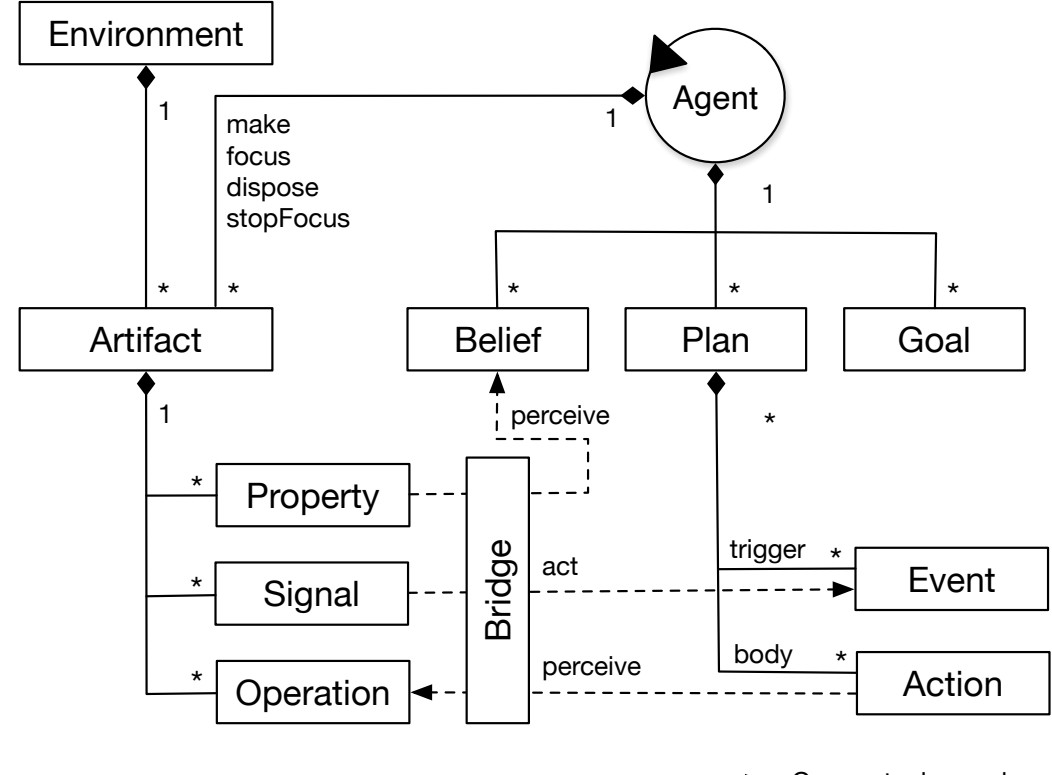

**Figure 1   The A&A meta-model, adapted from *Ricci et al. (2009)*.**

**Table 2  List of symbols used in the A&A meta-model and in this work.**

| Symbol | Description | Symbol | Description | Symbol | Description |
|---|---|---|---|---|---|
| Env | Environment | art | Artifact | id | Unique identifier |
| Pr | Set of artifact properties | Sg | Set of artifact signals | Op | Set of artifact operations |
| ag | Agent | bs | Set of beliefs | b | Believe |
| at | Predicate | t | Term | ps | Plan library |
| p | Plan | te | Triggering event | ct | Plan context |
| h | Plan body | a | Action | gs | Set of goals |
| g | Goal | u | Belief update | C | Agent circumstance |
| s | Agent state | I | Set of intentions | E | Set of events |
| A | Set of actions | Brg | Bridge interface | AO | Actions to operations mapping |
| F | Artifacts focused by agent | nid | Namespace identifier | | |

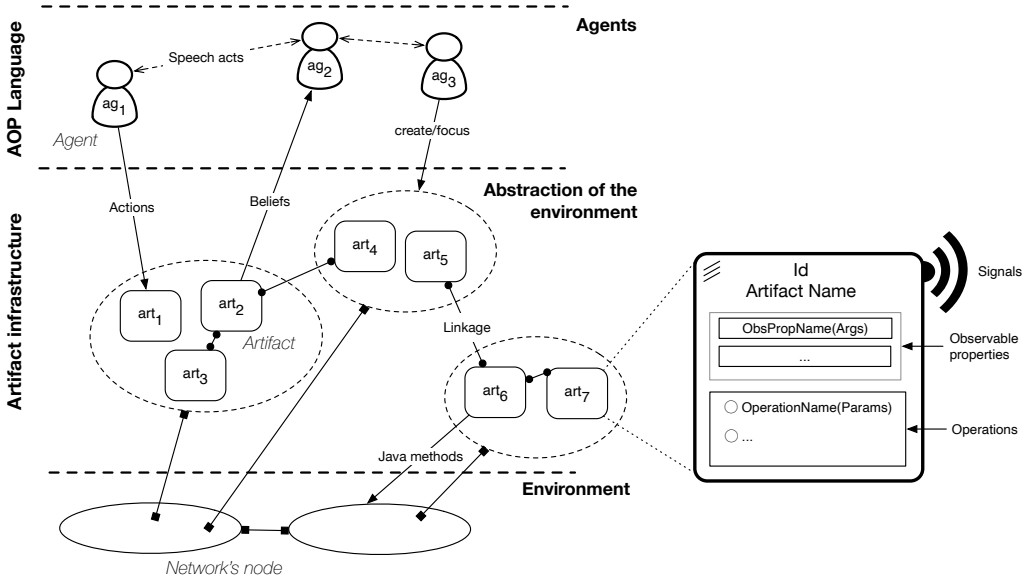

**Figure 2  A MAS under the A&A meta-model, adapted from *Ricci et al. (2009)*.**

and the artifacts, translating actions and events. A detailed description of artifacts, agents, and the bridge is provided in the next subsection.

## Artifacts

Artifacts, as shown in Fig. 2, are tools that provide operations able of being executed as actions by agents. When focusing on an artifact, an agent can perceive its observable properties and its emitted signals. Observable properties can be seen as state variables of the artifacts, that agents perceive as persistent beliefs. Signals are produced when executing operations, being perceived as non persistent beliefs.

The notation to denote the components of the artifact infrastructure is described as follows:

- Environment. A set of artifacts, $Env = \{art_1, \ldots, art_n\}$.
- Artifact. Each artifact is a tuple $\langle id, Pr, Sg, Op \rangle$, composed of:

  - A unique identifier $id$, provided by the agent that created the artifact. It is used to denote the artifact when used by the agents.
  - A set of observable properties $Pr = \{p_1, \ldots, p_n\}$;
  - A set of signals $Sg = \{s_1, \ldots, s_n\}$;
  - A set of operations $Op = \{o_1, \ldots, o_n\}$.

- Auxiliary functions. A set of functions to access the components of the artifacts are assumed, *e.g.*, $id(art)$ returns the identifier of the artifact $art$; $art(id)$ returns the artifact denoted by the identifier $id$; and so on.

## Agents

Although any AOP language can be adopted in the A&A meta-model, usually a BDI model of agency is assumed. This kind of models have strong philosophical foundations on three aspects of Intentionality, allowing for: (i) Representations based on the intentional stance (*Dennett, 1987, reprinted 2002*), where the behavior of the agents is modeled and interpreted in terms of beliefs, desires, intentions, and other intentional attitudes; (ii) reasoning methods based on the principles of practical reasoning (*Bratman, 1987*), where intentions are seen as plans; and (iii) communication based on speech acts (*Searle, 1969*). This model has been formally studied under different logics (*van der Hoek & Wooldridge, 2012*; *Meyer, Broersen & Herzig, 2015*) and implemented in different AOP languages (*Bordini et al., 2005*; *Bordini et al., 2006*).

Then, a particular concept is the *agent program* that can be described through an alphabet consisting of a finite set of symbols for variables (*Var*), constants (*Const*), functions (*Func*), predicates (*Pred*), and actions (*Actn*). The syntax of *ag*, defined in Table 3, includes:

- Beliefs (*bs*). Beliefs are a possibly empty set of ground first-order atoms, as those used in Prolog to represent facts. An atom (*at*) is a predicate applied to a certain number of terms. A term is a variable, a constant, or a function applied to a certain number of terms.
- Plans (*ps*). A non empty set of plans is assumed. Each plan $p$ has a trigger event *te* expressing for which event the plan is relevant. Trigger events include adding or deleting a belief, and adding or deleting a goal; a context *ct* expressing the conditions that make the plan applicable, as a logical formula; and a body $h$. The body is conformed by a sequence of actions (*a*), goals (*g*), and updates (*u*), *i.e.*, belief addition and deletion.
- Goals (*gs*). The agent program can include initial goals (*gs*), although this is not mandatory. Achievable goals (*!at*) are solved through practical reasoning, forming intentions with the plans of the agent to solve the goal. Test goals (*?at*) are solved through logical consequence from the beliefs of the agent.

We also focus on the components of the *AgentSpeak(L)* BDI model (*Rao, 1996*), that are relevant for modularization. This model is based on the concept of agent configuration, denoted by a tuple $Ag = \langle ag, C, s \rangle$, where $ag = \langle bs, ps, gs \rangle$ is an *agent program* composed

**Table 3** The syntax of an agent program $ag$, defined in BNF notation.

| | | | |
|---|---|---|---|
| $ag$ | ::= | $bs \quad ps \quad gs$ | |
| $bs$ | ::= | $b_1 \ldots b_n \mid \top$ | $(n \geq 1)$ |
| $b$ | ::= | $at$ | $(ground(at))$ |
| $at$ | ::= | $p \mid p(t_1, \ldots, t_n)$ | $(p \in Pred, n \geq 1)$ |
| $t$ | ::= | $v \mid c \mid f(t_1, \ldots, t_n)$ | $(v \in Var, c \in Const, f \in Func, n \geq 1)$ |
| $ps$ | ::= | $p_1 \ldots p_n$ | $(n \geq 1)$ |
| $p$ | ::= | $te : ct \leftarrow h$ | |
| $te$ | ::= | $+at \mid -at \mid +g \mid -g$ | |
| $ct$ | ::= | $ct_1 \mid \top$ | |
| $ct_1$ | ::= | $at \mid \neg at \mid ct_1 \wedge ct_1$ | |
| $h$ | ::= | $h_1; \top \mid \top$ | |
| $h_1$ | ::= | $a \mid g \mid u \mid h_1; h_1$ | |
| $a$ | ::= | $a \mid a(t_1, \ldots, t_n)$ | $(a \in Actn, n \geq 1)$ |
| $gs$ | ::= | $g_1 \ldots g_n \mid \top$ | $(n \geq 1)$ |
| $g$ | ::= | $!at \mid ?at$ | |
| $u$ | ::= | $+b \mid -at$ | |

of beliefs $ag_{bs}$, plans $ag_{ps}$, and goals $ag_{gs}$; and the *circumstance* of an agent $C = \langle I, E, A \rangle$ includes the intentions of the agent ($C_I$), the events perceived by the agent($C_E$), and the actions to be executed ($C_A$). The label $s$ indicates the state of the agent in the reasoning cycle induced by the operational semantics of the model.

The reasoning cycle of the agent is formally presented as a state transition system defined in *Bordini, Hübner & Wooldridge (2007)*, where the agent's configuration changes accordingly with a set of transition rules described as follows: The transition system of the operational semantics of AgentSpeak(L) includes *ProcMsg* for updating of those events $C_E$ perceived by agents; *SelEv* selects from $C_E$ one event to be processed. If there are no events, *SelInt* proceeds; *RelPl* computes the set of *relevantplans*, *i.e.,* the subset of $ag_{ps}$ useful to cope with the selected event. If there are no *relevantplans*, *SelEv* proceeds again; *ApplPl* computes the set of *applicableplans*, *i.e.,* the subset of *relevantplans* that can be executed accordingly to $ag_{bs}$. If there are no *applicableplans*, *SelInt* proceeds; *SelAppl* selects an *applicableplan*; *AddIM* adds the selected *applicableplan* to $C_I$; *SelInt* selects one intention from $C_I$ to be executed. If there are no intentions, *ProcMsg* proceeds; *ExecInt* executes the selected intention, *i.e.,* takes the plan in the top of the intention and processes the first element of its body. If this is an action, it is added to $C_A$; Finally, *ClrInt* clears $C$ as pertinent.

The transition system of the operational semantics is shown in Fig. 3.

## Bridge

Given the generic nature of the A&A meta-model, different models and implementations for the AOP language and the agent infrastructure can be adopted. It is necessary to define a bridge between them, for mapping the actions of the agents to the operations of the artifacts; and the observable properties and signals of the artifacts to the corresponding events and beliefs of the agents. The bridge can be defined as tuple $Brg = \langle E, AO, F \rangle$, where:

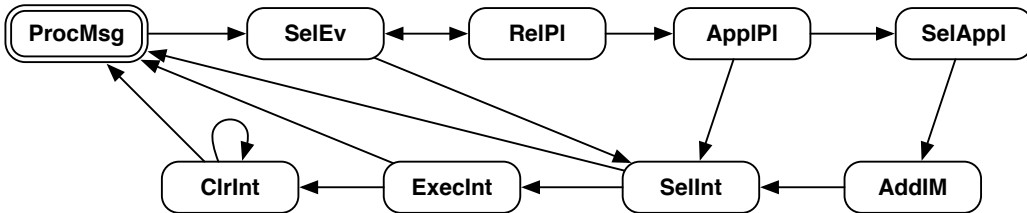

**Figure 3 Transition system of the operational semantics of AgentSpeak (L).** It describes the full agent reasoning cycle from processing events to executing intentions, adapted from *Bordini, Hübner & Wooldridge (2007)*.

- The observable events $E = \{e_1, \ldots, e_n\}$ is a set of observable properties and signals generated by the artifacts. Each event $e_i = \langle id, p|s, type \rangle$ is composed by the identifier of the artifact generating the event ($id$); the content of the event can be an observable property ($p$) or a signal ($s$); and the type of event. Three types of events can be generated by the artifacts: emitting a signal; and adding and removing an observable property.
- A mapping from actions to operations is required. The function $AO(ac) = op \mid \exists\, art \in Env \wedge ac = op \in op(art)$ returns the operation $op$ of an artifact $art$ in the environment $Env$, corresponding the action $ac$ executed by an agent.
- A focus register $F = \langle (ag, Art), \ldots \rangle$ storing couples denoting which artifacts $Art$ are focused by an agent $ag$.

Additionally, a set of functions to access the components of the bridge are assumed, *e.g.,* *focus*(*ag*) returns the set of artifacts being focused by the agent *ag*; *type*(*e*) returns the type of the event *e*; and so on. The bridge is an *ad hoc* component, strongly dependent of the AOP language and artifacts infrastructure. Different definitions are possible for the bridge, as long as they provide the functionality described here.

## MODULARIZATION IN THE AGENTS AND ARTIFACTS META-MODEL

Modularization in the A&A meta-model is enabled through the extension shown in Fig. 4. An agent program is now an aggregate of modules, resulting from loading at least an initial module. Each module has the same components of an agent program, *i.e.,* beliefs, plans, and goals, but every component is now associated with a *namespace*.

### Namespaces

The syntax proposed for the modules (see Table 4) adds a namespace declaration (*ns*), which is a list of atoms, possibly empty, of the form *namespace*(*nid*, *scp*), where *nid* is a namespace identifier and *scp* ∈ {*local*, *global*} indicates the scope of the namespace. A set of namespaces *Nid* is assumed. The proposed syntax allows the members of the alphabet (*Var* ∪ *Const* ∪ *Func* ∪ *Actn*) to be prefixed with a namespace identifier, and because of they account for beliefs, goals and actions, such prefixes allow these components to be associated with a defined namespace. Plans are also associated with namespaces due to their triggering events depend on beliefs or goals.

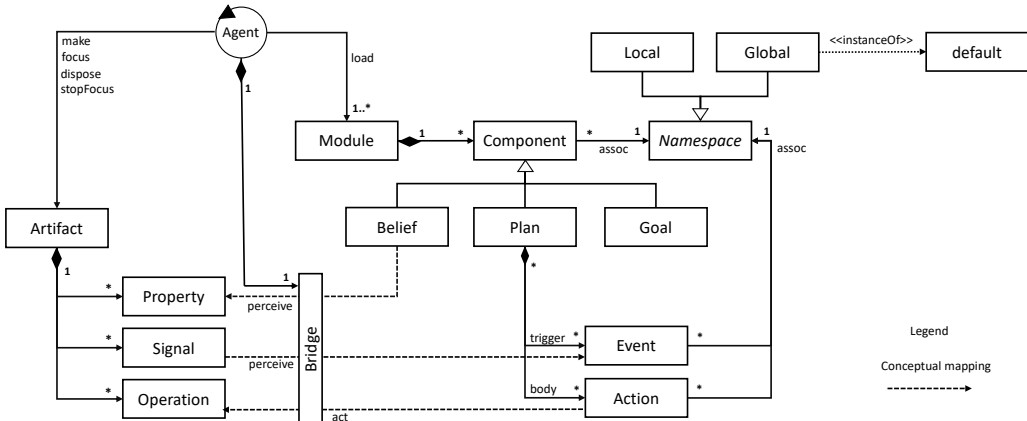

**Figure 4** The A&A meta-model extended for supporting modularization.

**Table 4** The syntax of a module, defined in BNF notation.

| | | | |
|---|---|---|---|
| $mod$ | ::= | $ns$  $bs$  $ps$  $gs$ | |
| $ns$ | ::= | $n_1,\ldots,n_j \mid \top$ | $(j \geq 1)$ |
| $n$ | ::= | $namespace(nid, scp)$ | $(nid \in Nid, scp \in \{global, local\})$ |
| $bs$ | ::= | $b_1 \ldots b_n \mid \top$ | $(n \geq 1)$ |
| $b$ | ::= | $at$ | $(ground(at))$ |
| $at$ | ::= | $p \mid p(t_1,\ldots,t_n)$ | $(p \in Pred, n \geq 1)$ |
| | \| | $nid :: p \mid nid :: p(t_1,\ldots,t_n)$ | $(nid :: p \in Pred, nid \in Nid, n \geq 1)$ |
| $t$ | ::= | $v \mid c \mid f(t_1,\ldots,t_n)$ | $(v \in Var, c \in Const, f \in Func, n \geq 1)$ |
| | \| | $nid :: v \mid nid :: c \mid nid :: f(t_1,\ldots,t_n)$ | $(nid :: v \in Var, nid :: c \in Const, nid :: f \in Func,)$ |
| | | | $(nid \in Nid, n \geq 1)$ |
| $ps$ | ::= | $p_1 \ldots p_n$ | $(n \geq 1)$ |
| $p$ | ::= | $te : ct \leftarrow h$ | |
| $te$ | ::= | $+at \mid -at \mid +g \mid -g$ | |
| $ct$ | ::= | $ct_1 \mid \top$ | |
| $ct_1$ | ::= | $at \mid \neg at \mid ct_1 \wedge ct_1$ | |
| $h$ | ::= | $h_1; \top \mid \top$ | |
| $h_1$ | ::= | $a \mid g \mid u \mid h_1; h_1$ | |
| $a$ | ::= | $a \mid a(t_1,\ldots,t_n)$ | $(a \in Actn \geq 1)$ |
| $a$ | ::= | $nid :: a \mid nid :: a(t_1,\ldots,t_n)$ | $(nid :: a \in Actn \geq 1)$ |
| $gs$ | ::= | $g_1 \ldots g_n \mid \top$ | $(n \geq 1)$ |
| $g$ | ::= | $!at \mid ?at$ | |
| $u$ | ::= | $+b \mid -at$ | |

Therefore, every component of a module may, or may not, have the namespace identifier as a prefix. For instance, the beliefs `price(50)`, `fixTV::price(50)`, and `fixPC::price(50)` are all different, since they are associated with different namespaces, as denoted by their prefixes. The same strategy is followed for other components of the modules, *e.g.*, plans, goals, actions. In this way, namespaces provide a syntactic solution

to the name-collision problem. The fact that the first belief in the previous example is not explicitly prefixed, means its namespace is abstract. Sometimes the namespace of a component is intended to be defined at run-time, after loading the module. In such case, it is said that the component has an abstract namespace, and it is not prefixed. Explicit prefixes indicate concrete namespaces. Abstract and concrete namespaces allow loading modules using different concrete namespaces, enabling a behavior close to the class/instance relation in OOP. In this way, namespaces contribute to enhance abstraction and hierarchization.

A namespace can be either local or global. Components associated with global namespaces can be accessed anywhere in the agent program. Those associated with a local namespace are only accessible in the module declaring the namespace. In this way, namespaces provide a syntactic solution to the information hiding principle and interfacing. Some auxiliary functions to work with namespaces include:

- $ids(mod) = \{id \mid id \in \{mod_{Var} \cup mod_{Const} \cup mod_{Func} \cup mod_{Actn}\}\}$, returns the set of all the identifiers used in the definition of the module $mod$.
- $ns(id) = ns$ returns the namespace of an identifier $ns :: id$.
- $local(id) = true$, iff $ns(id) = ns \wedge namespace(ns, local) \in mod_{ns}$, otherwise it is $false$.
- $global(id) = \neg local(id)$.
- $abstract(id) = true$, iff $ns(id) = \top$, otherwise it is $false$.

## Loading modules

In this proposal, an agent program $ag$ is the result of loading at least one module, which is not expressible at the syntactic level. The extended meta-model proposes a loading action for the modules. When a module is loaded, all its components associated with the abstract namespace are concretized. For this, a concrete namespace identifier ($nid$) must be specified when loading a module. Then, the module components are added to the agent program components. The following semantic rule, defines the loading operation as an action of the agent:

$$(\text{Load}) \frac{\mathcal{S}(C_A) = \texttt{load}(mod, nid)}{\langle ag, C, ExecInt \rangle \rightarrow \langle ag', C', ClearInt \rangle}$$

$$\begin{aligned}
\text{where:} \quad & mod' = mangling(mod, nid) \\
& ag'_{bs} = ag_{bs} \cup mod'_{bs} \\
& ag'_{ps} = ag_{ps} \cup mod'_{ps} \\
& ag'_{gs} = ag_{gs} \cup mod'_{gs}
\end{aligned}$$

The function $\mathcal{S}(C_A)$ denotes the action that has been chosen to be executed by the agent. The mangling function decorates the code of the loaded module accordingly to Algorithm 1. Basically, it replaces all the abstract namespaces in the loaded module $mod$, with the concrete namespace $nid$; it also decorates every local namespace with an internally auto-generated prefix denoted by # to make local namespaces inaccessible from other modules: Since the $id$ becomes syntactically incorrect for the parser, but still valid for the interpreter, it can only be used locally by the mangled module. For instance, the belief `priv::price(99)` is replaced by `#priv::price(99)`, if `priv` is declared as a local

namespace. Since `#priv::price` is a syntactically invalid identifier, no developer can code the access to this belief; it is only accessible in the mangled module declaring its namespace.

---

**Algorithm 1:** *mangling*(*mod*, *nid*) replaces every abstract namespace in *mod* with the concrete namespace *nid*; and decorates every local namespace with a prefix #, to make them inaccessible from other modules.

---

1 **begin**
       **Input**: *mod*: a module
       **Input**: *nid*: a concrete namespace
2     *mod′* = parse(mod)
3     **foreach** *id* ∈ *ids*(*mod′*) **do**
4         **if** *abstract*(*id*) **then**
5             replace *id* by *nid*::*id*
6         **if** *local*(*id*) **then**
7             replace *id* by #*nid*::*id*
8     **return** *mod'*

---

Loading the initial module of an agent is a special case. The first thing an agent program does is loading its initial module, assuming *nid* = *default*. The initial module program dispose for the agent its initial belief base and plan library and its set of goals. All components of the agent are empty before initialization.

Observe that, since the agent program is initialized loading a module, the namespaces provide an interface between two modules when loading. Figure 5 illustrates this: When a module $mod_1$ loads a module $mod_2$, interactions are bidirectional. The components of both modules associated with local namespaces are kept separate in each module. The components of $mod_2$, associated with an abstract namespace, are imported by $mod_1$, concretizing their abstract namespace; the concrete namespace, provided by $mod_1$, is used to extend $mod_2$. The components of $mod_2$, associated with a concrete global namespace, are imported by $mod_1$; those of $mod_1$ extends $mod_2$. This enables a relation close to the OOP, where the loaded module $mod_2$ source code can be seen as a template, and each load as an instantiation referred by the concrete namespace provided at loading time. This allows dynamically extending the functionality of modules in two ways. (i) by adding components to the concrete namespace used to load a module (instantiate), it is possible to extend the functionality for only such module instance; and (ii) to extend the functionality effecting all the instances of the same module, by adding components to the global namespaces of such module. In this way, namespaces provide a syntactic approach for interfacing.

Although *load* is a semantic rule, it is indeed a case of the rule for executing actions defined in *AgentSpeak(L)*. The result of executing *load* is strictly syntactical, an agent program *ag* decorated with namespaces. This means that the semantics of *AgentSpeak(L)* and its interpreter do not require further intervention.

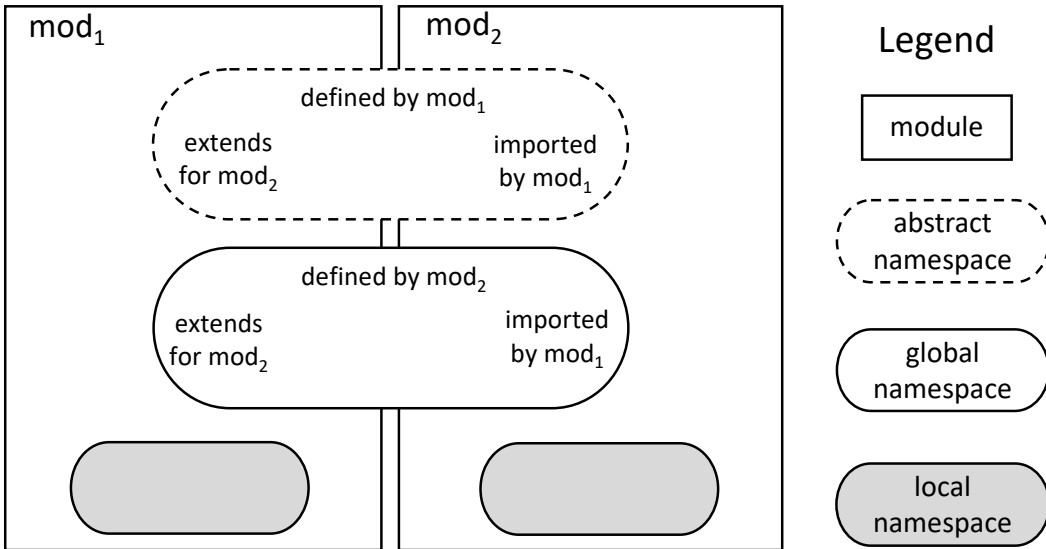

**Figure 5 The interface when module $mod_1$ loads the module $mod_2$.**

## Artifacts and namespaces

Namespaces provide a syntactic solution for integrating the perceptions and the actions of agents into modules. For this, the bridge ($Brg$) needs to be extended to support namespaces. The bridge includes:

- A focus register $F = \langle (ag, Art), \dots \rangle$ storing couples denoting which artifacts $Art$ are focused by an agent $ag$. However, in this version of the register, each $art \in Art$ has the form $(art, Nid)$, expressing that the artifact $art$ is focused by the agent using namespaces $Nid$.
- A set of functions to access the components of the focus register are assumed, *e.g.*, $focus(ag, art) = Nid$ returns the namespaces used by agent $ag$ to focus on artifact $art$.

This simple syntactic arrangement, allows a definition of modularization able of actions for creating, focusing, perceiving, using, and disposing artifacts. In what follows, the transition rules for these actions will be defined. Observe that these are transitions between states of a MAS under the A&A meta-model, *i.e.,* states have the form $\langle Ags, Brg, Env \rangle$. Most of the time, a single agent perspective is adopted, replacing the set of agents in the system $Ags$, with a single agent configuration $Ag = \langle ag, C, s \rangle$.

### *Creating artifacts*

Creating artifacts does not involve namespaces. An agent can create an artifact $art_{id}$ of type $artType$, by executing the action makeArtifact, provided by the environment ($Env$). We use function $makeArtifact/2$ to denote the low-level process of instantiating an artifact. A unique identifier ($id$) generated through the function $newUniqueId()$ is used to refer the artifact, which is added to the environment:

$$(\text{MakeArt}) \frac{\mathcal{S}(C_A) = makeArtifact(artType)}{\langle \langle ag, C, ExecInt \rangle, Brg, Env \rangle \rightarrow \langle \langle ag, C, ClearInt \rangle, Brg, Env' \rangle}$$

where:  $id = newUniqueId()$
$art = makeArtifact(id, artType)$
$Env' = Env \cup \{art\}$

### *Focusing on artifacts*

An agent starts perceiving an artifact $art \in Env$, by executing the action `focus`, provided by the environment. In this approach, a namespace must be provided when focusing:

$$(\text{Focus}) \frac{\mathcal{S}(C_A) = nid :: focus(art)}{\langle\langle ag, C, ExecInt\rangle, \langle E, AO, F\rangle, Env\rangle \rightarrow \langle\langle ag', C', ClearInt\rangle, \langle E, AO, F'\rangle, Env\rangle}$$

where:  $F' = F \cup \{(ag, (art, \{nid\}))\}$
$ag'_{bs} = ag_{bs} \cup \{nid :: p \mid p \in pr(art)\}$
$C'_E = C_E \cup \{+nid :: p \mid p \in pr(art)\}$

For example, suppose an agent focus on an artifact `auctionTool` with an associated namespace `auc1`. If the artifact includes an observable property `currentBid(9)`, then as a result of focusing, the belief `auc1::currentBid(9)` is added to the beliefs of the agent. A corresponding event to the belief addition is placed in the agent's events queue too. Observe that an agent can focus on the same artifact with different associated namespaces, replicating perception from an artifact in multiple modules.

### *Observable events*

The artifacts produce observable events that are perceived by the agents focusing on them. This means that some events in $C_E$ comes from the bridge, generated by some artifact in the environment. Since there are three types of observable events, the same number of transition rules are required. Observe that these "actions" are not actually executed intentionally by the agent. Observe that they are executed at $s = ProcMsg$ to update the events and beliefs of the agent. They are as follows:

- Signals. The events generated from artifact signals are also associated with a corresponding namespace. Signals from artifacts are placed as events in every namespace used to focus the artifact. For instance, a signal `tick` produced from artifact `tool`, focused by some agent using namespaces `ns1` and `ns2`, produces events `+ns1::tick` and `+ns2::tick` to be processed by the reasoning cycle of the agent. Since signals stand for non persistent events, the agent's belief base is not updated:

$$(\text{Signal}) \frac{\mathcal{S}(E) = \langle id, s, signal \rangle}{\langle\langle ag, C, ProcMsg\rangle, \langle E, AO, F\rangle, Env\rangle \rightarrow \langle\langle ag, C', ProcMsg\rangle, \langle E', AO, F\rangle, Env\rangle}$$

where:  $C'_E = C_E \cup \{+nid :: s\} \quad - \quad nid \in focus(ag, art(id))\}$
$E' = E \setminus \{\langle id, s, signal \rangle\}$

- Observable property addition. This type of percept is produced when an observable property is added in some artifact being observed by the agent. The observable property is added to the agent's belief base, in every namespace used to focus the artifact. An event reflecting this is added too. The percept is removed after being processed:

$$(\text{ObsPropAdded}) \frac{\mathcal{S}(E) = \langle id, p, obsPropAdded \rangle}{\langle \langle ag, C, ProcMsg \rangle, \langle E, AO, F \rangle, Env \rangle \rightarrow \langle \langle ag', C', ProcMsg \rangle, \langle E', AO, F \rangle, Env \rangle}$$

where:    $ag'_{bs} \; ag_{bs} \cup \{nid :: p\} \mid nid \in focus(ag, art(id))\}$

$C'_E \; C_E \cup \{+nid :: p\} \mid nid \in focus(ag, art(id))\}$

$E' \; E \setminus \{\langle id, p, obsPropAdded \rangle\}$

- Observable property deletion. When an observable property has been deleted from some artifact being focused, such observable property is removed from every namespace used to focus the artifact. An event to notify the belief deletion is generated, and the percept is removed:

$$(\text{ObsPropDel}) \frac{\mathcal{S}(E) = \langle id, p, obsPropDel \rangle}{\langle \langle ag, C, ProcMsg \rangle, \langle E, AO, F \rangle, Env \rangle \rightarrow \langle \langle ag, C', ProcMsg \rangle, \langle E', AO, F \rangle, Env \rangle}$$

where:    $ag'_{bs} = ag_{bs} \setminus \{nid :: p\} \mid nid \in focus(ag, art(id))\}$

$C'_E = C_E \cup \{-nid :: p\} \mid nid \in focus(ag, art(id))\}$

$E' = E \setminus \{\langle id, p, obsPropDel \rangle\}$

Observable property updates can be handled as a sequence of rules (**ObsPropAdded**) and (**ObsPropDel**). However, at implementation level a fourth type of observable event, standing for an artifact observable property update, is considered for practical reasons.

### Stop focusing artifacts

Agents can stop focusing artifacts by executing the action `stopFocus`. This causes that the percepts and operations of a specific artifact cease to be mapped to the agent:

$$(\text{StopFocus}) \frac{\mathcal{S}(C_A) = nid :: StopFocus(id)}{\langle \langle ag, C, ProcMsg \rangle, \langle E, AO, F \rangle, Env \rangle \rightarrow \langle \langle ag', C', ProcMsg \rangle, \langle E, AO, F' \rangle, Env \rangle}$$

where: $F' = F \setminus \{(ag, (art(id), nid))\}$

$ag'_{bs} = ag_{bs} \setminus \{nid :: p \mid p \in pr(art(id))\}$

$C'_E = C_E \cup \{-nid :: p \mid p \in pr(art(id))\}$

### Artifacts disposal

Artifact disposal complements the artifact creation. When an artifact is disposed, every belief and action related to the artifact is removed, and the artifact is not longer eligible to be focused, because it is also removed from the environment. Events to notify and reflect belief deletions are produced to be processed if necessary. Agents can use the action `dispose` to eliminate artifacts:

$$(\text{Dispose}) \frac{\mathcal{S}(C_A) = nid :: dispose(id)}{\langle \langle ag, C, ProcMsg \rangle, \langle E, AO, F \rangle, Env \rangle \rightarrow \langle \langle ag', C', ProcMsg \rangle, \langle E, AO, F' \rangle, Env' \rangle}$$

where:    $F' = F \setminus \{(ag, (art(id), nid))\}$

$ag'_{bs} = ag_{bs} \setminus \{nid :: p \mid p \in pr(art(id))\}$

$C'_E = C_E \cup \{-nid :: p \mid p \in pr(art(id))\}$

$Env' = Env \setminus \{art(id)\}$

### *Operation execution*

The execution of an action *ac* associated with some namespace denoted by *nid*, is processed as follows:

$$(\text{ExecAct}) \frac{\mathcal{S}(C_A) = nid :: ac \wedge OA(ac) = op \wedge nid \in focus(ag, art(op)) \wedge exec(op)}{\langle\langle ag, C, ProcMsg\rangle, \langle E, AO, F\rangle, Env\rangle \rightarrow \langle\langle ag, C, ProcMsg\rangle, \langle E, AO, F\rangle, Env\rangle}$$

where: $C'_A = C_A \setminus \{nid :: ac\}$

The rule results in matching an action *ac* with a corresponding operation *op*, with the restriction that the namespace *nid* associated with action *ac* must be currently used to focus on the artifact implementing *op*. In other words, the action *ac* is successfully executed only if a corresponding operation *op* to *ac* is available in namespace *nid*. The execution of operation is carried on by function *exec*(…). Finally, it must be highlighted that default artifact operations, *e.g.*, `makeArtifact`, `focus`, etc., are available in any namespace.

## IMPLEMENTATION

To evaluate our proposal, it has been implemented in JaCa –the integration of Jason *Bordini, Hübner & Wooldridge (2007)*, an extended interpreter for AgentSpeak(L) *Rao (1996)*, implemented in Java; and CArtAgO *Ricci et al. (2009)*, an infrastructure grounded on the A&A meta-model, also implemented in that language. Some general implementation issues are discussed here, before introducing a case study in the next section. The implementation consists basically in extending Jason for allowing namespaces and providing two ways of loading modules; and extending the bridge with CArtAgO for allowing the artifacts to exploit namespaces too. Both extensions are merely syntactical, *i.e.,* the interpreter of Jason and the CArtAgO implementation do not change, warranting in this way, full backward compatibility with non-modular MAS written in JaCa. The implementation as described here is part of the official distribution of Jason 2.0, and both the platform and source code are available online (*Hübner & Bordini, 2016*).

Jason is extended exploiting its facilities to adopt user defined pre-processing directives. A pre-processing directive `namespace/2` is added for allowing the declaration of namespaces. The first argument of the directive is the identifier of the namespace, and the second one is its scope (local or global). The identifiers of the expressions enclosed within this directive are associated with the specified namespace. Other identifiers can be associated with the namespace by prefixing them with the corresponding namespace identifier.

The pre-processing directive `include`, and the internal action `.include`, are extended to take a second parameter for supporting namespaces. Both implement the rule **load** defined in Section 'Loading Modules'. The name of file containing the source code of the module is their first argument, and the second one is its associated namespace. If a free variable is passed as the second argument of `.include`, an unique namespace identifier is internally auto-generated, it is bounded to the variable, and then, it is used to load the module. The internal action and directive can be used for dynamically and statically loading modules, respectively.

For practical reasons, the identifiers used for terms are associated with the default namespace when a module is loaded, as long as they are not explicitly prefixed.

However, a prefix `::` in a term denotes its association with the abstract namespace, *e.g.*, `instartedAuct(::tool)`, the term `tool` is related to the abstract namespace. In Jason, the keywords, strings, lists, and numbers terms cannot be associated with namespaces. Jason defines two types of actions: The environment is modified by an external action, on the contrary, an internal action is fired internally to the agent. Internal actions cannot be associated with namespaces, while external actions can.

The bridge between Jason and CArtAgO is also customized to deal with namespaces. Jason implements this bridge as an agent architecture class, defining how percepts and actions are handled. This class is extended to implement the rules introduced in Section 'Artifacts and Namespaces'.

Artifacts are instances of the Java class `Artifact`, provided by CArtAgO. They can be denoted by an identifier, as defined in Section 'Artifacts', and by a name defined by the programmer. Names are implemented as a Jason term, allowing then the use of namespaces to avoid name clashes when using them in modules. If a namespace is not explicitly specified when creating an artifact, the *default* namespace is adopted, because terms are associated with such namespace by default.

Observe that the operations provided by the artifacts are invoked by the agents through their external actions. These actions can be annotated to indicate the artifact preferred by the agent when executing an action. Such annotations have precedence over the namespace prefix. The predefined operations, excepting `focus` and `focusWhenAvailable`, which have a different semantic regarding namespaces, omit the prefix because they are available in every module.

## CASE STUDY

This section offers a detailed portrait of our proposal, presenting a MAS for running auctions to contract services. The complete JaCa project source code of the case study is available for download at *Ortíz-Hernández (2020a)*. Additionally, a more complex example of the contract-net-protocol (both modular and non-modular versions) is distributed with the Jason 2.1 release (*Hübner & Bordini, 2020*).

The modules `auction` and `participant` (Codes 6 and 7) encapsulate the functionality to create and taking part of an auction, respectively. The MAS compounds bob and `alice`, the auctioneer agents, whose initial modules are showed in Codes 2 and 3 respectively; and the participants `companyA` and `companyB` (Codes 4 and 5). In this implementation, every instance of module `auction` isolates the beliefs and events of each negotiation, so that they do not interfere with other negotiations or even other module components. Our case study shows how the percepts and actions from observed artifacts are independently handled by multiple modules for their own purposes.

Artifacts of type `AuctArt` (Code 1) are used by auctioneers and participants to manage their auctions. These artifacts are created and initialized by the auctioneers, using a task description and the initial price as parameter, *e.g.*, line 9 of Code 6. They provide an operation `bid` for placing an offer, *e.g.*, lines 19–26; and an operation `close` for closing an auction. When an auction finalizes the corresponding artifact signals the final winner

(c.f. line 16 of Code 1). Agents focusing the artifact can perceive through the artifact's observable properties and signals, the task being auctioned, current bid, current winner and the final winner. Our case study shows how the percepts and actions from observed artifacts are independently handled by multiple modules for their own purposes.

```java
public class AuctArt extends Artifact {

  private boolean open=true;

  public void init(String taskDesc, double maxPrice)  {
    defineObsProperty("task",          taskDesc);
    defineObsProperty("currentBid",    maxPrice);
    defineObsProperty("currentWinner", "no_winner");
  }

  @OPERATION
  public void close() {
    ObsProperty opCurrentWinner = getObsProperty("currentWinner");
    if(getOpUserName().equals(getCreatorId().getAgentName()))
      open=false;
      signal("won", opCurrentWinner.stringValue());
  }

  @OPERATION
  public void bid(double bidValue) {
    ObsProperty opCurrentValue  = getObsProperty("currentBid");
    ObsProperty opCurrentWinner = getObsProperty("currentWinner");
    if (open & bidValue < opCurrentValue.doubleValue()) {
      opCurrentValue.updateValue(bidValue);
      opCurrentWinner.updateValue(getOpUserName());
    }
  }
}
```

Code 1: AuctArt.java

```
// an initial goal
!contract([site,wall,floor,roof,pool]).

+!contract([])
  <- .wait(2500);  // a deadline
     // close auctions and notify winners
     for(A::instanceOf(auction)){
        !!A::close
     }.

+!contract([T|Ts])
  <- // var A is bounded to a unique nsp id
     // one instance for each auction
     .include("auction.asl",A);
     // starts auction for task T,
     // no maximum price
     // !! posts a sub-goal with new focus
     !!A::start(T,9999);
     !contract(Ts).
```

Code 2: bob.asl

```
{include("logger.asl",log)} // static load

{begin namespace(priv,local)}
  max_price(floor,1200).
  max_price(plumbing,600).
  max_price(pool,1000).
{end}

!contract([floor,plumbing,pool]).

+!contract([])
  <- .wait(4500);
     for(A::instanceOf(auction)){
        A::close
     }.
+!contract([T|Ts]): priv::max_price(T,P)
```

```
17    <- .include("auction.asl",A);
18       !!A::start(T,P);
19       log::focusWhenAvailable(A::tool)
20       !contract(Ts).
```

Code 3: alice.asl

```
 1 {include("participant.asl",p)}
 2
 3 my_price(1650). // minimum plus~150
 4
 5 // acceptable tasks
 6 p::accept([site,wall,roof]).
 7
 8 p::trust([alice]). // trusted agents
 9
10 // I can improve a bid if
11 p::improve(Bid,T,Bid-150):- my_price(P) & P <= Bid.
```

Code 4: companyA.asl

```
 1 {include("participant.asl",p)}
 2
 3 my_name("company_B").
 4
 5 overworked:- my_name(Me)
 6    & .count(A::currentWinner(Me),C) & C >= 3.
 7
 8 p::acceptable(_).
 9 p::trust(All):- .all_names(All).
10 p::improve(Bid,_,math.random(999)+800)
11   :- not overworked.
```

Code 5: companyB.asl

```
 1 instanceOf(auction).
 2
 3 {namespace(priv,local)} // forward definition
 4
 5 @p1[atomic]
 6 +!start(Task,MaxPrice)
 7    <- //:: associates a term with abstract nsp
 8       makeArtifact(::tool,
 9       "jcm.AuctArt",[Task,MaxPrice],Aid);
10       +priv::state(open);
11       .broadcast(tell,::started(Task));
12       // percepts from Aid go to abstract nsp
13       focus(Aid).
14
15 // operator -+ updates beliefs
16 @p2 +!close <- -+priv::state(closed); close.
17
18 // allow other modules get the state of auction
19 @p3 +?state(S) <- ?priv::state(S).
20 @p4 -?state(none).
```

Code 6: auction.asl

```
 1 acceptable(T):- accept(Ts) & .member(T,Ts).
 2 trusted(Ag):- trust(Ags)  & .member(Ag,Ags).
 3
 4 @p[atomic]
 5 +A::started(Task)[source(Ag)]
 6  : trusted(Ag) & acceptable(Task)
 7    <- .include("bidder.asl",A);
 8       +{A::improve(Bid,MyBid)
 9         :- improve(Bid,Task,MyBid)
10        };
11        // used here to focus by name
12        A::focusWhenAvailable(A::tool).
13
14 // auction artifact produces a signal
15 // to notify the final winner,
```

```
16  // react if it is me!
17
18  +A::won(MeS)
19   : .my_name(Me) & .term2string(Me,MeS)
20      <- println("I Won Auction",A,"!").
```

Code 7: participant.asl

The agent bob creates multiple auctions. It applies the internal action `.include/2` for dynamically loading module `auction` (line 14) and to start an auction for each element in a defined list of tasks (initial goal at line 2). It uses dynamic definition of namespaces (second parameter of action include is a variable) to encapsulate the internal state of each auction it creates, such that each namespace used to load the module can be seen as an auction instance from the auctioneer's point of view. Posting the sub-goal `start` in namespace *A* launches the auction (line 18). Next, the agent bob waits until a fixed deadline has expired to close all auctions (lines 4–9). The agents participating in those auctions can observe the corresponding artifacts to find out the final winners.

Agent `alice` runs auctions for building a pool and setting up the floor and plumbing of its house (line 9). It statically defines local namespace `priv` for hiding its maximum price for each task from other modules (line 3). The identifiers without an explicit namespace between lines 4 and 6 will be placed in the local namespace `priv`. Artifacts are created at line 9 of Code 6, and the beliefs, events and actions mapped from them are handled by the instances of module `auction`. The line 1 loads module `logger`, which provides functionality for informing the progress of specific auctions. A pre-processing directive `include/2` is used to perform a static loading of `logger`. Agent `alice` also places the percepts from the auction artifacts into namespace `log` (line 19; in this way, the events produced from them can be independently used by module `logger` for printing out information about the progress of auctions.

Dynamic loading and namespaces allow `bob` and `alice` to encapsulate the components related to a particular auction, *e.g.*, observe how auctioneer agents can initialize an auction using an identical name for a task, without producing a name-collision.

The `companyA` is interested in auctions for tasks `site`, `wall` and `roof`. It participates only in auctions executed by `alice`. When an auction starts under these conditions, the directive `.include/2` is used for loading the module `participant` (line 1). This agent extends the functionality of the module instance by adding beliefs to the namespace where the module was loaded (p); namely, it adds beliefs about tasks of interest and trusted agents (lines 6 and 8). The agent sets up its strategy for participating in auctions at line 11. The module `participant` uses the beliefs added to the namespace where it was loaded to decide what tasks can be accepted, what agents are trusted and to know how much and when to bid (cf. lines 6, 8 and 9 of Code 7). A different strategy is adopted by the `companyB`, who accepts all auctions but never commits to a new task if it is overworked, *i.e.,* it abstains from participating in an auction if it is already winning more than three (lines 5 and 6 of Code 5). The `companyB` bids a bounded random quantity (line 11).

The `auction` module encapsulates the functionality to create and manage an auction. It starts with a belief used to identify the module instances, specifically, those namespaces used as an instance of module `auction`, *e.g.*, lines 7 and 13 of Codes 2 and 3, respectively.

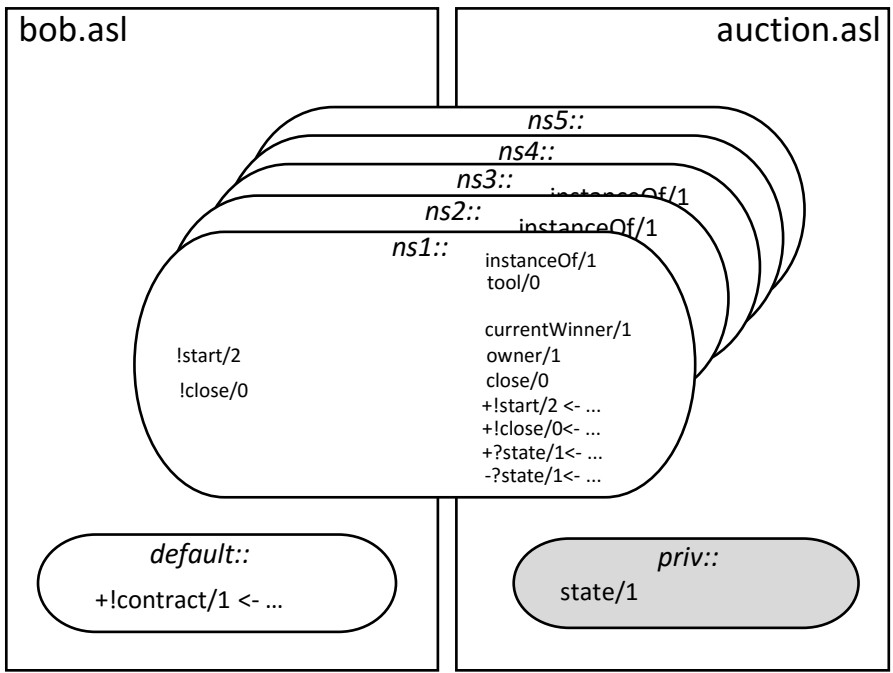

**Figure 6** **The namespaces of agent bob during its execution.**

Next, a forward declaration of the local namespace priv in line 3 is defined. Observe that the namespace of start/2 is abstract, and a concrete namespace will be given when the module is loaded (lines 14, 17 and 7 of Codes 2, 3 and 7, respectively). The auction module exports elements associated with the abstract namespace, *e.g.*, plans @p1 and @p2. The local namespace priv is not visible to other modules, thus it does not interfere or clash with any namespace in other modules, even if they have the same name, *e.g.*, line 3 of Code 3. Namespace priv in module auction is used to encapsulate the belief representing the current state of a running auction, it means that such belief is accessible only from this module, *e.g.*, line 10. However, a loader module can retrieve the current state of the auction by means of plans @p3 and @p4. The Fig. 6 illustrates the relation between the modules bob and auction using the same notation of Fig. 5 in Section 'Loading Modules'.

The module participant provides functionality for participating in auctions. The plan @p1 is executed if an auction has been started, and the participant is interested in the task and trusts the auctioneer agent. A part of the functionality needed for participating in auctions, is in turn modularized into a complementary module called bidder, which provides agents with the appropriated functionality for consulting if they are winning an auction, or not (c.f. line 1 of Code 8), as well as to place bids following their own strategy (lines 3 and 4 of Code 8). Module bidder is loaded at line 7, in such a way that each instance *A* of module bidder, represents an auction from the point of view of the participant. In lines 8 and 9, the participant module extends the instances of bidder to specify the strategy adopted for bidding in each auction. Next, at line 12, auction artifacts are focused in the same namespace used to load the module bidder, with the purpose of letting the

instances of such module apply its rules and plans to the corresponding perceptions and artifact operations. Finally, the plan defined in lines 18–20 is executed when the agent results the final winner of an auction.

```
1  i_am_winning:- .my_name(Me) & term2string(Me,MeS) & currentWinner(MeS).
2
3  +currentBid(Bid): improve(Bid,MyBid) & not i_am_winning
4    <- bid(MyBid).
```

Code 8: bidder.asl

The `logger` module implements the 'view' part of the auction system, illustrating separation of concerns. It prints in the console relevant information about active auctions. The behavior of informing the progress of specific auctions can be easily activated (and deactivated). Moreover, it can be independently developed and maintained from the modules implementing the main functionality of the system.

```
1  +task(T)
2    <- println("An auction for task",T,"has started!").
3
4  // reports all bids
5  +currentBid(V): task(T) & currentWinner(W)
6    <- println("The current winner of auction for task",T,"is",W,"with a bid of",V).
```

Code 9: logger.asl

## EVALUATION

With the purpose of evaluating the advantages of our approach, we also developed another version of the case study with no modularity. Consequently, we implemented six extensions to these versions. The source code of all modular and non-modular versions is available for download (*Ortíz-Hernández, 2020c*). The initial version is composed of two auctioneer agents (`alice` and `bob`) and two participants who are `companyA` and `companyB`. The first extension consists in modifying agent *alice* behavior to extend their deadline in case that some of its auctions has resulted without a winner. The second and third versions add participant agents with individual bidding strategies into the MAS (*i.e.,* `companyC`, `companyD`, `companyE` and `companyF`). In the fourth, auctioneer agents can optionally print-out the progress of negotiations for specific auctions (as illustrated through module `logger` presented in Section 'Case Study'). The fifth provides agents with functionality for registering statistical information about determined auctions. Finally, in the sixth we generalize the MAS to let `alice` bid any set of tasks instead of only those related to building a house; so that fixed plans for implementing the workflow of building a house are removed, and tasks are delegated to winners immediately after the auction is closed, expecting they have a proper plan to perform such task.

The comparison between the versions is shown in Tables 5 and 6. Both coupling and cohesion metrics are inspired from strategies proposed in *Jordan & Collier (2012)* and *García-Magariño, Cossentino & Seidita (2010)*. Coupling is estimated by summing every single use of a component by external modules, and cohesion is calculated by counting the number of connected components in the internal dependencies graph of each module similarly as the 4th method of calculating the lack in cohesion of methods (LCOM4). The scripts used to estimate coupling and cohesion scores are available at *Ortíz-Hernández (2020b)*.

**Table 5  Contrasting the Auction MAS over a series of extensions.** (*Files*) specifies the number of source code files that compose the MAS; (*Rewrites*) the working components that had to be reedited; (*Length*), (*Difficulty*) and (*Effort*) correspond to the measures from the Halstead complexity metrics (*Halstead, 1977*); (*Updates*) summarizes the block additions and deletions of source code that had to be performed over existing components, counted in basis of a *diff* algorithm; (*m*) stands for modular version; and (*n*) for non-modular version.

| # | Extension Description | Files | | Rewrites | | Length | | Difficulty | | Effort | | Updates | |
|---|---|---|---|---|---|---|---|---|---|---|---|---|---|
| | | *m* | *n* | *m* | *n* | *m* | *n* | *m* | *n* | *m* | *n* | *m* | *n* |
| | Starting implementation | 10 | 9 | 0 | 0 | 1210 | 1377 | 33.14 | 37.80 | 1788 | 2322 | 0 | 0 |
| 1 | Wait until all auctions have a winner | 10 | 9 | 1 | 1 | 1371 | 1614 | 34.19 | 38.88 | 2091 | 2798 | 14 | 21 |
| 2 | Add two participants (scale) | 12 | 11 | 1 | 1 | 1703 | 1957 | 40.34 | 45.82 | 3108 | 4056 | 2 | 2 |
| 3 | Another two participants | 14 | 13 | 1 | 1 | 1908 | 2152 | 44.81 | 49.71 | 3868 | 4839 | 2 | 2 |
| 4 | Print-out progress of auction | 15 | 13 | 1 | 2 | 2031 | 2276 | 45.21 | 50.85 | 4154 | 5236 | 2 | 10 |
| 5 | Register auction statistics | 16 | 14 | 1 | 2 | 2327 | 2609 | 49.38 | 52.27 | 5268 | 6169 | 10 | 34 |
| 6 | Delegate tasks to winners | 16 | 14 | 1 | 2 | 2181 | 2591 | 46.45 | 51.44 | 4645 | 6029 | 10 | 21 |

**Table 6  Comparing the Auction MAS through a series of extensions.** (*Plans*) specifies the number of plans in the MAS; (*Components*) stands for the sum of all goals, beliefs, events, actions, *etc.* in the modules composing the MAS; (*Coupling*) the overall level of coupling of MAS; (*Cohesion*) the average of the cohesion scores of modules, where value 1 means that module fulfills the principle of cohesion, and higher values suggest that the module should be split. (Uncoupled) and (Coupled) denote the number of uncoupled and coupled Components, respectively; (*m*) stands for modular version, *i.e.*, coded under the proposed approach; and (*n*) for the version without modularization.

| # | Extension description | Plans | | Components | | Coupling | | Cohesion | | Uncoupled | | Coupled | |
|---|---|---|---|---|---|---|---|---|---|---|---|---|---|
| | | *m* | *n* | *m* | *n* | *m* | *n* | *m* | *n* | *m* | *n* | *m* | *n* |
| | Starting implementation | 7 | 8 | 140 | 149 | 422 | 616 | 1.50 | 1.0 | 28 | 33 | 112 | 116 |
| 1 | Wait until all auctions have a winner | 7 | 8 | 150 | 163 | 470 | 752 | 1.50 | 1.0 | 28 | 32 | 122 | 131 |
| 2 | Add two participants (scale) | 9 | 10 | 196 | 210 | 876 | 1247 | 1.50 | 1.0 | 32 | 34 | 164 | 176 |
| 3 | Another two participants | 11 | 12 | 225 | 240 | 1284 | 1681 | 1.50 | 1.0 | 31 | 33 | 194 | 207 |
| 4 | Print-out progress of auction | 11 | 13 | 237 | 249 | 1357 | 1830 | 1.42 | 1.0 | 31 | 30 | 206 | 219 |
| 5 | Register auction statistics | 12 | 14 | 270 | 285 | 1736 | 2468 | 1.25 | 1.0 | 29 | 27 | 241 | 258 |
| 6 | Delegate tasks to winners | 12 | 14 | 262 | 287 | 1651 | 2499 | 1.25 | 1.0 | 32 | 31 | 230 | 256 |

The extensions are progressive; therefore, updates are counted as the necessary modifications to achieve the current extension from the previous one. Extensions two and three are not extensions but escalations; for the sake of clarity we will homogenize as extensions all the system updates. Because of the available space, the effort measure has been multiplied by $10^{-2}$ and rounded. The Total row shows the increases from the initial implementation to the last version (sixth); and for the rewrites and updates columns, it sums the times that a component had to be rewritten and the updates performed to existing code, respectively.

For example, to fulfil the 5th extension of the modular version (we began with the 4th one), we performed ten updates (code additions and deletions) in one file, which increased the length, effort and difficulty measures of the system programs in 14.5%, 9.2% and 26.8% respectively (*i.e.,* 296 points more of length, 4.17 of more difficulty and 1,114 of extra effort), when compared with the 4th extension (the previous one). Analogously, to extend the non-modular version, one file was edited to perform deletions and additions over twenty-one blocks of code, increasing the program length in 14.6% (333 points),

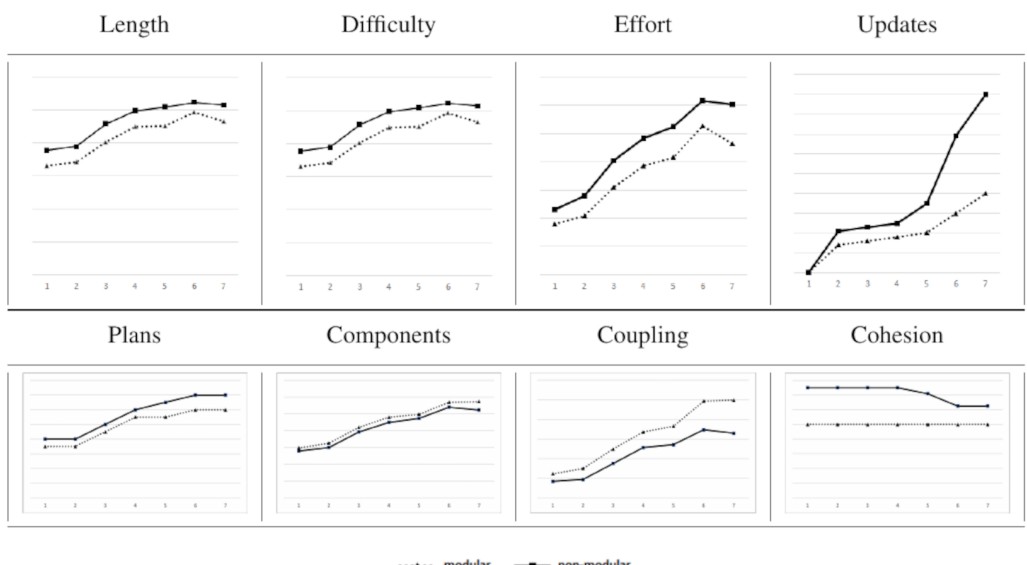

**Figure 7  Scores of the auction MAS's according to the Halstead metric for complexity (Length, Difficulty, and Effort); where *X*-axis denotes the version of the MAS from the initial implementation (1) to that resulting of performing all six extensions (7) (c.f. Table 5), and Y-axis corresponds to the score.** The updates chart summarizes the block additions and deletions of source code that had to be performed to implement each extension. The plans and components charts stand for the total plans and components (*i.e.,* beliefs, goals, events, *etc.*), respectively. The coupling chart shows the sum of coupling scores of all modules in the MAS. The cohesion chart presents the average cohesion per module, the maximum value for this score is 1, and higher values mean less cohesive modules.

the difficulty in 2.8% (1.42 points) and the effort measure by 17.8% (933 points), when compared with its previous extension. One source file was added to the system in both versions, as result of the extension.

According to the results, a total of 40 updates and six rewrites were needed in the modular version. On the contrary, in the non-modular version 90 updates and nine rewrites were required. Particularly, in this case study we decreased the duties of maintainability and extensibility by 55.5% and 33.3%, respectively (*i.e.,* 50 updates and three rewrites less). Therefore, we can argue that our approach makes easy the development of a project in terms of maintenance and extensibility.

The Halstead metric scores better the modular versions in all its complexity measures, *i.e.,* length, difficulty and effort (c.f. Figure 7, lower is preferred). By analyzing this particular case study, it is because in the non-modular version, extra terms are needed in multiple occasions for disambiguating which components (*e.g.,* beliefs, plans, goals) are referred by identifiers. This increases both vocabulary and length of the system programs. Specifically in JaCa, by integrating namespaces with artifacts following our solution, the use of annotations `artifact_name` and `artifact_id` (c.f. *Ricci, Piunti & Viroli (2011)*) is considerably reduced.

From extensions one to three, where participant agents are added to the MAS, both versions needed the same number of updates and suffer a similar increase of Halstead

measures. This can be explained because Jason already provides an `include/1` directive which allows incorporating code from a source file, and the functionality needed to participate in auctions was developed in a separate file which is included by the participant agents at booting; similarly to including a module but without namespaces support. However, it worth emphasizing that the old Jason directive (`include/1`) does not solve the name collision issue and it also lacks information on hiding support. It just avoids the code repetition across multiple files. To illustrate this consider the auctioneer agent bob using `price/2` (*e.g.*, to set its maximum price), when it loads the source file with the implementation of the functionality to participate in an auction (by means of an include with no namespaces support), due to belief `price/2` is already used by the including file to calculate the amount of bids for tasks, it ends in a name-collision. Then, the produced behaviour is not clearly determined (Madden and Logan *Madden & Logan (2010)* report on this, based on the usage of the `include` directive incorporated in earlier Jason releases for building Multi-Agent System of large-scale *Madden & Logan (2007)*). To cope with this, it is necessary either to rename the belief used by bob for setting its maximum price, or perform the name change in the included file. If the latter option is intended to be use, updating every source file to avoid symbol duplicity is needed.

From extension four to six the modular versions take advantage in the score. This is because of the mechanism that allows focusing artifacts in multiple namespaces, *e.g.*, to make an observable event, such as `close` signal, trigger multiple plans in different modules, such that each module follows a particular and independent course of action (plan) as result of the same percept (*e.g.*, for printing-out information, registering statistics and notifying winners of auctions to delegate the corresponding task). To achieve extensions four to six in the modular version, the current working code remained almost unchanged, since a new module was written each time that was needed to independently handle the specific event with a new plan. Contrastingly, in the non-modular version, *i.e.*, without namespaces, the exclusive plan to handle the event had to be rewritten each time that it was required to trigger an additional course of action for a particular event; considerably increasing the number of updates.

The results also show that, as long as the system grows through extensions, the difference of the scores between both versions increases progressively, which suggest that after multiple extensions the non-modular version will be even more complex, hence harder to maintain and extend than the modular version (c.f. Figure 7).

The following section provides an overview of our proposal for modules pointing out the topics mentioned in the 'Introduction'; it remarks the main properties of our proposal contrasting them with the related work discussed in 'Related Work'. It also presents a discussion about how some Object-Oriented programming concepts can be modeled in Agent-Oriented programming using our namespace-based solution for modularization.

## DISCUSSION

BDI-AOP languages benefit from the notion of namespace to address modularization. Under this approach the main concerns of modularization can be overcome: (a) the

association of each component with a unique namespace solves the name-collision problem: reference disambiguation is possible due the usage of qualified names; (b) the interface comprises the global namespace concept, to make possible both importing components and extending module's functionality; (c) dynamic association of components of a module to namespaces is granted by the notion of abstract namespace, this allows composing a solution with higher degree of complexity by two possibilities: loading multiple times the same module in disparate namespaces, and loading several modules into one namespace. (d) encapsulation of components is possible thanks to local namespaces making easy the development of modules in an independent way, thus modules are committed to handle just a given set of percepts and actions requested by programmers. Loading modules at runtime results in the acquisition of new capabilities on the fly (without execution interruption), so this can be seen as a kind of dynamic updating. The strategy we use to perform modularization is the key difference of our approach. First, adding more information to the identifiers of components, allows establishing a logical way to organize them in the mental state of agents. Second, those approaches described in 'Related Work' commonly require introducing additional transition rules in order to handle several belief bases and plan libraries, even event queues, in one reasoning cycle, *i.e.,* they consider modules as active components within the operational semantics due to they deal with multiple mental states inside agents. As a consequence, the implementation of solutions is more difficult. On the other hand, our approach provides a solution at the syntax level. Thus, to implement our proposal in different BDI languages it is enough just to perform a parser extension.

Next, considering what *Nunes (2014)* discusses and some principles from Object-Oriented programming, we argue about the suitability of our approach to set up different relationships between modules, namely: association, composition and generalization. Note that capability is a quite similar concept of module, both include a set of beliefs, plans and goals (*Busetta et al., 1999*; *Braubach, Pokahr & Lamersdorf, 2006*). Consequently, those relationships related to capabilities also apply to our notion of module.

## Association

With the execution of any plan of a *loader* module a goal is required, and the plan to achieve it belongs to the *loaded* module, then we say there is an association. According to *Nunes (2014)*, one consequence of association is to increase cohesion because of functionality modularization allows addressing different concerns using separate modules.

In Codes 10 and 11 an example of association is provided: the module one loads the A module for the execution of one plan from it. The information hiding principle holds when the module is loaded by a local namespace, *i.e.,* in this relationship module A ignores about module one.

```
1  {namespace(ia,local)}
2
3  +!do <- .include("A.asl",ia);
4        !ia::inc(2).
```

Code 10: one.asl

```
1  count(0).
2
3  +!inc(S): count(X)
4     <- -+count(X+S).
```

Code 11: A.asl

## Composition

Compared with the association, relationship composition is a stronger one. According to *Nunes (2014)* there are situations where components belonging to the *loader* module are used by the *loaded* module. Despite the increase in coupling between modules, containment notion can be modeled granting information hiding. Such a relationship is not implemented in our approach, allowing preserving the information hiding principle and to reduce module coupling. We opted to use arguments to share information between modules *load* and *loaded* whenever they are required. However, the modeling of the composition relationship described by *Nunes (2014)* is possible with namespaces: it suffices with the addition of a symbol to refer the abstract namespace of the loader module. An example of this can be seen in Codes 12 and 13. A module named B get access to one belief from module two: rate/1. Then, when the plan do/0 from module two the output is the printing of counter 1. The symbol ° is used here to indicate the abstract namespace of *loader*. To support this, an extension of the mangling function (c.f. algorithm 1) can be performed in order to substitute the ° symbol by the right namespace at loading-time.

```
1  {namespace(ib,local)}
2
3  rate(0.50).
4
5  +!do <- .include("B.asl",ib);
6           !ib::inc(2);
7           ?ib::count(X);
8           .print("counter ",X).
```

Code 12: two.asl

```
1  count(0).
2
3  +!inc(S): count(X)
4     <- ?°::rate(R);
5        -+count(X+S*R).
6
7
8
```

Code 13: B.asl

## Cardinality

Cardinality in module associations can be represented due to the fact that one module can be loaded into several namespaces while instances keep its own beliefs. For instance, in Codes 3 and 7 from the 'Case Study' section, the module bob loads an initiator instance for each contract net protocol it fires, then, the state of each negotiation is preserved.

### Visibility

With local namespaces, components can be kept private within a module. On the other hand, global namespaces can be used to share components between all modules. Moreover,

when sharing components among the instances of the same modules is required, for instance to avoid the replication of the same information over and over, our approach allows representing this by proposing a new level for namespace visibility (along with *global* and *local*). In this way, a *module* namespace grant access to all its instances only. This is similar to the concept of class visibility level in the Object-Oriented Programming approach, for example, the Java modifier `static`.

### Multi-Inheritance

Our approach makes it possible to implement a concept like this one through the union of modules. It is possible to reuse beliefs, plans and goals from multiple modules, to compose new modules implementing a more complex and specialized behavior. The example shown in Codes 14 and 15 reproduces the case when a module C inherits B by placing together all its components in one namespace (line 9). When a local namespace has a *parent*, it is hidden to the *child* module, and vice-versa. A parent module can be included at the end of the source code, this *overrides* the current plan `inc/1` in A (if multiple plans are applicable for an specific event, the default selection function chooses the first plan added to the agent's plans library). This latter works particularly well in Jason given the fact that the first plan within the code is the selected one for execution, no matter if there are several applicable plans to handle an event.

*Dhaon & Collier (2014)* present a more sophisticated solution for languages alike AgentSpeak(L). Their method customizes the selection function that the interpreter uses to determine the next plan to be executed, so a disambiguation of plans to be executed is performed whenever different implementations of the same plan occurs at different level within the hierarchy of modules.

At runtime level, a dynamic extension of modules is also achieved by using the namespaces notion. For instance, in Code 7 at lines 8–12, the functionality of module `auction` (c.f. Code 6) is extended. This can be convenient in the case that an extension of the functionality of only one instance is required but there is no need to create a new module.

Related to this topic, it is worth mentioning a solution introduced by *Baldoni, Giordano & Martelli (1995)*, that seems more general for modeling multi-inheritance in agent-oriented programming. Such approach is grounded in the scope of logic programming, and consists not in the use of namespaces but a modal operator, to group rules; thus the inheritance relationship between modules is defined by a set of logic implications. However, its feasibility for being adopted in the context of agent-oriented programming should be carefully analyzed and evaluated.

```
1  {namespace(ic,local)}
2
3  +!init
4     <- .include("C.asl",ic);
5        !ic::inc(2);
6        !ic::mult(2);
7        ?ic::count(X);
8        .print("counter " X).
9
```

Code 14: three.asl

```
1  //belief count/1 is inherited from A
2  +!mult(T): count(X)
3     <- -+count(X*T).
4
5  //overrides plan inc/1 in A
6  +!inc(S): count(X)
7     <- -+count(X+1).
8
9  {include("A.asl")}
```

Code 15: C.asl

# CONCLUSION AND FUTURE WORK

We proposed a solution for BDI Agent development pinned to the A&A meta-model following the principles of modularization; we also explored the premise that the notions of namespace and bridge, as proposed in this article, are suitable to cope with the main problems related to modularization, namely, overcoming name-collisions, maintaining the information hiding principle, offering an interface, and integrating environment components with modules. We illustrated some examples of the properties and feasibility of our approach by implementing it under the framework of JaCa. Furthermore, the present approach was successfully implemented as part of the official distribution of Jason (*Bordini, Hübner & Wooldridge, 2007*) and CArtAgO (*Ricci et al., 2009*), a widely used platform for programming MAS.

As a future work, we aim to provide a mechanism to unload modules for allowing removing all components from modules that agent is no longer using. It is also relevant to consider and analyze the design efforts at early phase of development related to tasks of decomposing functionality into modules and defining its interface, as well as explore a refactoring algorithm to automatically perform modularization. We have to point out that the design effort is close related to the complexity of the intended task, where refactoring is really needed. In our work, we established the basis to make this possible. Finally, we aim to explore the use of namespaces to modularize organizational reasoning plans and patterns, *e.g.*, as conceived by Moise (*Hannoun et al., 2000*), to implement our proposal in other languages and test multiple study cases to deeply evaluate the generality of the discussed approach.

## Funding

The authors received no funding for this work.

## Competing Interests

The authors declare there are no competing interests.

## Author Contributions

- Gustavo Ortiz-Hernndez conceived and designed the experiments, performed the experiments, analyzed the data, performed the computation work, prepared figures and/or tables, authored or reviewed drafts of the article, and approved the final draft.

- Alejandro Guerra-Hernndez conceived and designed the experiments, authored or reviewed drafts of the article, and approved the final draft.
- Jomi F. Hbner conceived and designed the experiments, performed the computation work, authored or reviewed drafts of the article, and approved the final draft.
- Wulfrano Arturo Luna-Ramrez conceived and designed the experiments, analyzed the data, prepared figures and/or tables, authored or reviewed drafts of the article, and approved the final draft.

## Data Availability

The code is available in the Supplemental Files.

The Jason 2.1 Code that includes the implementation of our proposal is also available at SourceForge: https://sourceforge.net/projects/jason/files/jason/version%202.1. (Code authored by Jomi Hbner (Department of Automation and Systems, Federal University of Santa Catarina, Florianpolis, SC, Brazil), Rafael Bordini (Pontifcia Universidade Catlica do Rio Grande do Sul, Porto Alegre, RS, Bazil))

The Java Code (scripts to automatically analyse the metrics to evaluate the case of study) and the Jason Project of our case of study are also available at SourceForge: https://sourceforge.net/projects/house-building. (Code authored by Gustavo Ortiz-Hernandez).

## Supplemental Information

Supplemental information for this article can be found online at http://dx.doi.org/10.7717/peerj-cs.1162#supplemental-information.

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
