# Peer review of "Modularization in Belief-Desire-Intention agent programming and artifact-based environments"

_PeerJ Computer Science, doi:10.7717/peerj-cs.1162_

## Round 0.1 · original submission · Major Revisions

Based on reviewers’ comments, you may resubmit the revised manuscript for further consideration. Please consider the reviewers’ comments carefully and submit a list of responses to the comments along with the revised manuscript.

Reviewer 1 ·

Basic reporting

This paper proposes an extension for the Agents and Artifacts meta-model to enable modularization. The Belief-Desire-Intention (BDI) model of the agency was employed, and then modules are intended to encapsulate functionality into independent and reusable units of code, expressed in BDI terms, which can be dynamically loaded by agents. The practical feasibility of the proposal is demonstrated by developing an auction scenario, where the source code enhances scores of coupling, cohesion and complexity metrics when compared against the non-modular version of the system.

A. The literature review should highlight the weakness of the existing work. It should be in chronological order. For example, in line 138 where a paper from 2013 is presented followed by a paper dated 2008 on line 144. Similarly, table 1 should be sorted w.r.t. the presented date.
B. Figure 1 is missing.
C. If the Algorithm given in line 210 is in the form of pseudo then it will be a bit easier to understand
D. Some very basic mistakes e.g. table caption should be on the top of the table but here is the opposite (see tables 1,2).
E. Repetition of contents is found in the document in some places. For example, Tables 4 and 5 are explained in the text as well as in the table legend. This type of text is somehow self-explanatory. So, it can be removed.
F. Enough references have been cited.
G. The guidelines provided by PeerJ for manuscript uploading are not followed. For example, tables, figures etc. should be uploaded separately and removed from the actual write-up.

Experimental design

Enough experimental design details are provided to replicate the work.

Validity of the findings

Different figures are given but how results are generated to draw these figures are missing. Similarly, some important details are missing e.g. figure 7, x-axis and y-axis labels are missing.

Additional comments

The paper has everything needed it is only missing some minor things which need to be addressed.

Reviewer 2 ·

Basic reporting

In this article the authors have proposed an extension of Agents and Artifacts meta-model, they have adopted the BDI model of agency, there are couple of major points to be considered by the authors. Following are my recommendations
The abstract is not very clear, eg. The reader has difficulty to find the adopted methodology of the proposed extended model as well as the improvements with regards to the exiting model which has been extended by the authors.
It seems that the related work section needs to be improved with state of the art variations of the modularization models, as in the current literature survey, the authors have used many old modules to discuss. Please find some latest published studies and implement your study accordingly.
In section 3 the Figure 1. The A&A meta-model, adapted from Ricci et al. (2009b). is missing , please cross check
The authors needs to give the reference of figure 2 in the captioned.
In this statement Environment. For the sake of simplicity, an environment is assumed to be a set of n g 1 artifacts,
184 Env = {art1, . . . ,artn}., how can you justify that the artifacts could only be one as indicated by the value of n, which is n>=1 ?
The experimentation and evaluation has been performed well, and I am satisfied with that. Just if you can elaborate it more with more than one case studies then it will be more justified to claim the accuracy and enhancement of your proposed model.

Experimental design

In this article the authors have proposed an extension of Agents and Artifacts meta-model, they have adopted the BDI model of agency, there are couple of major points to be considered by the authors. Following are my recommendations
The abstract is not very clear, eg. The reader has difficulty to find the adopted methodology of the proposed extended model as well as the improvements with regards to the exiting model which has been extended by the authors.
It seems that the related work section needs to be improved with state of the art variations of the modularization models, as in the current literature survey, the authors have used many old modules to discuss. Please find some latest published studies and implement your study accordingly.
In section 3 the Figure 1. The A&A meta-model, adapted from Ricci et al. (2009b). is missing , please cross check
The authors needs to give the reference of figure 2 in the captioned.
In this statement Environment. For the sake of simplicity, an environment is assumed to be a set of n g 1 artifacts,
184 Env = {art1, . . . ,artn}., how can you justify that the artifacts could only be one as indicated by the value of n, which is n>=1 ?
The experimentation and evaluation has been performed well, and I am satisfied with that. Just if you can elaborate it more with more than one case studies then it will be more justified to claim the accuracy and enhancement of your proposed model.

Validity of the findings

In this article the authors have proposed an extension of Agents and Artifacts meta-model, they have adopted the BDI model of agency, there are couple of major points to be considered by the authors. Following are my recommendations
The abstract is not very clear, eg. The reader has difficulty to find the adopted methodology of the proposed extended model as well as the improvements with regards to the exiting model which has been extended by the authors.
It seems that the related work section needs to be improved with state of the art variations of the modularization models, as in the current literature survey, the authors have used many old modules to discuss. Please find some latest published studies and implement your study accordingly.
In section 3 the Figure 1. The A&A meta-model, adapted from Ricci et al. (2009b). is missing , please cross check
The authors needs to give the reference of figure 2 in the captioned.
In this statement Environment. For the sake of simplicity, an environment is assumed to be a set of n g 1 artifacts,
184 Env = {art1, . . . ,artn}., how can you justify that the artifacts could only be one as indicated by the value of n, which is n>=1 ?
The experimentation and evaluation has been performed well, and I am satisfied with that. Just if you can elaborate it more with more than one case studies then it will be more justified to claim the accuracy and enhancement of your proposed model.

Additional comments

In this article the authors have proposed an extension of Agents and Artifacts meta-model, they have adopted the BDI model of agency, there are couple of major points to be considered by the authors. Following are my recommendations
The abstract is not very clear, eg. The reader has difficulty to find the adopted methodology of the proposed extended model as well as the improvements with regards to the exiting model which has been extended by the authors.
It seems that the related work section needs to be improved with state of the art variations of the modularization models, as in the current literature survey, the authors have used many old modules to discuss. Please find some latest published studies and implement your study accordingly.
In section 3 the Figure 1. The A&A meta-model, adapted from Ricci et al. (2009b). is missing , please cross check
The authors needs to give the reference of figure 2 in the captioned.
In this statement Environment. For the sake of simplicity, an environment is assumed to be a set of n g 1 artifacts,
184 Env = {art1, . . . ,artn}., how can you justify that the artifacts could only be one as indicated by the value of n, which is n>=1 ?
The experimentation and evaluation has been performed well, and I am satisfied with that. Just if you can elaborate it more with more than one case studies then it will be more justified to claim the accuracy and enhancement of your proposed model.

Reviewer 3 ·

Basic reporting

The topic of the paper is very interesting by adopting modularization for BDI Agent. However, I think the paper need still a further work to be published. I think that in short, the authors will achieve it and will have a nice publication.

1- The Abstract need to rewrite. The authors should further simplify the language. In the abstract the authors should talk about the domain of the article to try to fix. Also, some definition not well known so need to the reader brief knowledge about it such as namespace.
2- In the introduction section, firstly, the content is highly redundant, because the authors only need to review some related references and present own motivations. About some presentation of references, they are inaccurate and undetailed. The authors should compare and review related references and show what they have done in detail. For the motivations of this paper, the authors should give why they are motivated by tackling with this problem and what they aim to achieve as compared to what was done previously. The authors need to further clarify.
3- In general, the paper should proofread again using academic language.(i.e., As usual, a set of functions to access the components of the bridge are assumed,”) cannot use A usual in the article. Please make improve your writing to have strong publication.

4- The text in Fig 2 is not clear. I do recommend increasing the size of the text in this figure.

5- I do recommend adding a table showing the symbols, especially the math symbols (i.e., art1,…).

6- I recommend the authors to discuss the complexity of algorithm 1.

Experimental design

No comment

Validity of the findings

No comment

---

## Round 0.2 · accepted · Accept

The reviewers are satisfied with the current revision and your paper has been recommended for publication, congratulations.

Reviewer 1 ·

Basic reporting

Mentioned changes have been incorporated now.

Experimental design

Experimental design was already good.

Validity of the findings

no comment.

Additional comments

I think manuscript is good to go now.

Reviewer 2 ·

Basic reporting

my comments are addressed , no further comments

Experimental design

my comments are addressed , no further comments

Validity of the findings

my comments are addressed , no further comments

Additional comments

my comments are addressed , no further comments

Reviewer 3 ·

Basic reporting

I have no further comments. The authors address all my comments

Experimental design

Fair

Validity of the findings

Fair